# Characterization and Biological Activities of In Vitro Digested Olive Pomace Polyphenols Evaluated on Ex Vivo Human Immune Blood Cells

**DOI:** 10.3390/molecules28052122

**Published:** 2023-02-24

**Authors:** Claudio Alimenti, Mariacaterina Lianza, Fabiana Antognoni, Laura Giusti, Onelia Bistoni, Luigi Liotta, Cristina Angeloni, Giulio Lupidi, Daniela Beghelli

**Affiliations:** 1School of Biosciences and Veterinary Medicine, University of Camerino, 62032 Camerino, Italy; 2Department for Life Quality Studies, Alma Mater Studiorum, University of Bologna, 47921 Rimini, Italy; 3School of Pharmacy, University of Camerino, 62032 Camerino, Italy; 4Rheumatology Unit, Department of Medicine, University of Perugia, 06126 Perugia, Italy; 5Department of Veterinary Science, University of Messina, 98168 Messina, Italy

**Keywords:** olive pomace, PBMC, cytokines, immune gene expression, HPLC-DAD, antioxidants, inflammation, metabolome, phenolic bioaccessibility

## Abstract

Olive pomace (OP) represents one of the main by-products of olive oil production, which still contains high quantities of health-promoting bioactive compounds. In the present study, three batches of sun-dried OP were characterized for their profile in phenolic compounds (by HPLC-DAD) and in vitro antioxidant properties (ABTS, FRAP and DPPH assays) before (methanolic extracts) and after (aqueous extracts) their simulated in vitro digestion and dialysis. Phenolic profiles, and, accordingly, the antioxidant activities, showed significant differences among the three OP batches, and most compounds showed good bioaccessibility after simulated digestion. Based on these preliminary screenings, the best OP aqueous extract (OP-W) was further characterized for its peptide composition and subdivided into seven fractions (OP-F). The most promising OP-F (characterized for its metabolome) and OP-W samples were then assessed for their potential anti-inflammatory properties in ex vivo human peripheral mononuclear cells (PBMCs) triggered or not with lipopolysaccharide (LPS). The levels of 16 pro-and anti-inflammatory cytokines were measured in PBMC culture media by multiplex ELISA assay, whereas the gene expressions of interleukin-6 (IL-6), IL-10 and TNF-α were measured by real time RT-qPCR. Interestingly, OP-W and PO-F samples had a similar effect in reducing the expressions of IL-6 and TNF-α, but only OP-W was able to reduce the release of these inflammatory mediators, suggesting that the anti-inflammatory activity of OP-W is different from that of OP-F.

## 1. Introduction

Climate change, loss of biodiversity and environmental pollution increase are challenges that must be faced by improving the relationship between humans and ecosystems. With this aim, EU environmental policy and legislation strongly encourages the reuse and recycling of waste, a reduction in harmful chemicals, and the use of environmentally friendly compounds that are also technologically satisfactory and economically convenient.

Agri-food industries are among the principal producers of waste and by-products in the world [1]. The European Union alone produces about 90 million tons of food by-products every year, with an impressively negative effect on the environment [2]. For these reasons, researchers are paying more and more attention to these wastes not only as a potential source of energy, but also as a source of bioactive molecules. In recent years, agri-food by-products have been increasingly considered for the extraction of bioactive compounds such as antioxidants, vitamins, minerals, dietary fiber, essential fatty acids, oligosaccharides and oligopeptides [3,4]. In the Mediterranean area, a huge amount of waste is generated during the olive oil production process [5].

The Mediterranean basin contains approximately 98% of the planted olive (*Olea europea* L.) trees, and together with other European countries, produces 80% of the world’s olive oil [6]. Consequently, the olive oil industry generates significant amounts of olive oil by-products (olive pomace, olive leaves and olive mill wastewater), which need to be managed by these countries according to strategies aimed toward reducing the impact on the environment through the sustainable re-use of agri-food waste.

Olive pomace (OP) is an olive oil by-product rich in high-value compounds (e.g., polyphenols, dietary fiber, unsaturated fatty acids, antioxidants and minerals) and, in the context of a sustainable economy, the interest in recovering and utilizing bioactive compounds to add health benefits to the diet has increased during recent years [7].

Currently, there is indeed a wide bibliography in favor of the beneficial health effects of extra virgin olive oil, as well as on the possibility of obtaining valuable bioactive compounds from the waste products of the olive oil processing process (olive pomace and olive mill wastewater). A search in PubMed using the terms “olive oil AND health” or “olive by-products AND health” produced 3073 and 85 results, respectively (updated on 2 January 2023). Indeed, besides the well-recognized healthy effects of extra virgin olive oil [8,9,10], beneficial properties have also been demonstrated for processing by-products (leaves and olive mill wastewaters) including anti-cancer [11], prevention against age-related diseases [12], cardioprotective, anti-diabetic [13] and anti-inflammatory [14], among others [7,15,16,17]. In a recent study by Markhali et al. [18], oleuropein, one of the most common bioactive compounds in olive oil by-products, was found to be effectively capable of rebuilding the tissue damage caused by cisplatin in the stomach and the lungs, whereas Žugčić et al. showed that the compounds found in olive leaves exerted positive effects on gut microbiota [19].

The health-promoting effects of OP have mainly been associated with the presence of antioxidants, especially those belonging to plant-specialized metabolites, attributable to five classes of polyphenols (biophenols) identified as secoiridoids, simple phenols, flavonoids, phenolic acids, and lignans [20]. Interestingly, OP extract has been demonstrated to ameliorate lipid accumulation and lipid-dependent oxidative unbalance [21]. However, high amounts of α-tocopherol (2.63 mg/100 g) and fatty acids have also been identified as bioactive compounds in OP by Nunes et al. [22], and a relevant contribution to the reported beneficial effects of OP in preventing cardiovascular and gut diseases has been attributed not only to polyphenols, but also to sugars and minerals present in the pomace by Ribeiro et al. [23].

Di Nunzio et al. [24] demonstrated that an aqueous OP extract was able to significantly reduce IL-8 secretion, one of the main proinflammatory cytokines, in Caco-2 cells in both basal and inflamed conditions, suggesting OP as a potential low-cost, high added-value ingredient for the formulation of functional and innovative food [24,25].

With a view to a potential use of OP in the formulation of innovative and functional foods or nutraceuticals, in this study we evaluated the prospective anti-inflammatory properties of OP compounds following digestion in the gastrointestinal (GI) compartments and passing the mucosal and intestinal barriers. Indeed, it has been observed that the bioavailability of polyphenols greatly changes during digestion, due to their different degrees of absorption, stability, solubility, and permeability [15,23,26].

To this purpose, three batches of sun-dried OP were characterized for their profiles in phenolic compounds and in vitro antioxidant properties before (methanolic extracts) and after (aqueous extracts) their simulated in vitro digestion and dialysis. The most promising aqueous extract, selected based on its composition of bioactive compounds, was further characterized and tested for its anti-inflammatory potential using PBMC cells.

PBMC cells were chosen because, circulating in the blood stream throughout the body, they represent not only the first systemic cell lines acting in the innate and adaptative immune responses, but also one of the two most represented cell categories (leukocytes vs. red blood cells) of the first tissue in which these bioactive compounds enter the body.

## 2. Results

A preliminary characterization of the three different OP batches (OP1, OP2, and OP3) was carried out to identify the OP batch with the highest potential biological properties before and after in vitro digestion. Only the OP extract with the highest potential biological properties was used for the subsequent experiments.

### 2.1. Total Phenol Content (TPC) and Phenolic Characterization of OP Extracts

The total phenol content and the individual phenolic characterization of each OP extract obtained from the three different batches of OP was determined before (methanol extracts) and after (aqueous extracts) the simulated in vitro digestion.

#### 2.1.1. Total Phenol Content (TPC) and Phenolic Characterization of Methanolic OP Extracts

In addition to the evaluation of the total content of phenolic compounds (TPC), a targeted HPLC-DAD analysis was carried out on the three methanolic OP extracts to identify and quantify some of the characteristic compounds of OP belonging to different chemical classes, such as secoiridoids, catechols, diterpens, flavonoids, hydroxycinnamic and phenolic acids (Table 1). Significant differences in the phenolic content were found among the three extracts, with OP1 being the richest for most metabolites. The most abundant compound was luteolin; its highest concentration was found in OP1. The biggest differences among the extracts were found for hydroxytyrosol and tyrosol, which were in the ranges of 4.9–224.6 µg and 8.2–223.0 µg/g, respectively, with OP1 showing the maximum level, and OP3 the minimum level. Regarding hydroxycinnamic acids, caffeic and chlorogenic acids were detected in all extracts. OP2 showed the highest content of caffeic acid, while OP1 was the richest in chlorogenic acid. Gallic acid levels were below the limit of quantification (LOQ) in all samples. Considering the total targeted metabolite index (TTMI), which represents the sum of the identified compounds, its value was significantly higher in OP1 than both OP2 and OP3 (Table 1).

#### 2.1.2. Total Phenol Quantification (TPC) and Phenolic Characterization of Aqueous OP Extracts

Table 2 reports the phenolic compositions and TPC of the two types of aqueous extracts (< or >3.5 kDa) obtained from the three OP batches after in vitro digestion and dialysis.

The aqueous OP extracts, characterized by the presence of bioavailable compounds (serum available) with a molecular weight (m.w.) <3.5 kDa after dialysis [23] were indicated as OP-W (1, 2 or 3); whereas the non-available digested aqueous extracts were indicated as OP-W n.a. (1, 2 or 3; m.w. > 3.5 kDa).

The distribution of the detected metabolites in the bioavailable and non-bioavailable aqueous extracts obtained after the in vitro digestion varied, depending on the molecule type (Table 2). As a general trend, most metabolites detected in the pomace were also found in the absorbable fraction, with a different percentage of recovery, depending on samples. Only apigenin was not found, neither in the non-absorbable or the absorbable samples.

The bioaccessibility index, calculated as the percentage of the bioactive compound which was solubilised after the intestinal dialysis in reference to its total content in the undigested food, is reported in Table 3. A high bioaccessibility (more than 80%) was found for hydroxytyrosol and tyrosol in OP1 and OP2, while undetectable levels of both metabolites were present in OP3, probably due to their low levels in the original pomace extract (Table 2). Similar values were found for myricetin and caffeic acid, while the bioaccessibility of pinoresinol and chlorogenic acid were slightly lower, in the range of 46–70%. Oleuropein and ligstroside were found in the absorbable fraction at percentages ranging from 14 to 50% for the former, and 11 to 43% for the latter. A lower bioaccessibility was observed for luteolin, with a percentage of about 14%, with no differences among the three samples.

### 2.2. Antioxidant Properties and Reducing Power of OP Extracts

The antioxidant potential and reducing power of each OP extract obtained from the three different batches of OP before and after the simulated in vitro digestion were evaluated by three different spectrophotometric assays.

#### 2.2.1. Antioxidant Properties and Reducing Power (ABTS, DPPH and FRAP Assays) of Methanolic OP Extracts

Table 4 reports the antioxidant properties and reducing powers evaluated in the methanolic extracts obtained from the three crude OP batches (1, 2, and 3). In accordance with the different results obtained for the phenolic content and TTMI index, OP1 methanolic extracts showed the highest antioxidant activity of all the antioxidant assays utilized, whereas OP2 resulted in an intermediate position between the other two extracts. Therefore, the trends of antioxidant responses resembled the trends of TPC found in the three different OP batches.

#### 2.2.2. Antioxidant Properties and Reducing Power (ABTS, and FRAP Assays) of Aqueous OP-W n.a. and OP-W Extracts after In Vitro Digestion and Dialysis

Table 5 reports the results of the radical scavenging assays evaluated in the aqueous extracts obtained from the in vitro digestion and dialysis of the three OP batches (1, 2, and 3). All the digested OP samples conserved their own antioxidant properties proportionally to the content of the original bioactive compounds, therefore, the OP1 samples, both absorbable and not, showed the highest antioxidant activity when compared with the other two OP samples.

#### 2.2.3. Antioxidant Property (ABTS) and TPC of Aqueous OP-W Fractions (OP-F)

Based on the previous results, OP-W1 was selected as the most promising extract, and further fractionated using reverse-phase chromatography (HPLC-DAD) on a semipreparative C18 column. Figure 1 shows the seven different major chromatographic peaks obtained. All these peaks were characterized by low hydrophobicity, as suggested by the fact that they eluted at a low concentration of acetonitrile (10–20%).

The material was selectively eluted at each peak and then tested for its radical scavenging activity (ABTS assay) and TPC (Table 6). The fifth peak resulted as the most biologically active (*).

### 2.3. OP-W Peptide Identification and Possible Bioactivity

The peptide content of OP-W was detected by mass spectrometry. After digestion, we obtained 78 and 93, or 96 and 112 peptides (before and after further trypsinization) and some of them permitted the identification of 13 and 14 proteins specific to *Olea europaea* olive or to saprophytic microorganisms of the olive tree plant, respectively. Table 7 and Table 8 report the number of peptides identified in the OP-W1 samples, and the corresponding proteins were searched in both “Olea” and “Olea Europea” protein databases. Some of these proteins, such as 50S ribosomal protein L16, amine oxidase, pectinesterase, 2, profilin-1, 4-coumarate-CoA ligase, and putative geraniol 10 hydroxylase, as expected, were derived from *Olea europaea*; whereas, others from *Pseudomonas* (sp. PIC125 and PIC 141) belonged to bacteria with potential as a biocontrol tool against pathogenic microorganisms (i.e., Verticillium dahlia Kleb.) of olive plants [27].

The profile of these peptides was checked in the open-access tool PeptideRanker (a score higher than 0.6 was considered as potentially “bioactive”) to forecast the eventuality of biological activity of a peptide sequence [28], and two or five peptides of the *Olea europaea* olive received a score between 0.66 and 0.79, or 0.63 and 0.68, before and after trypsinization, respectively. These peptides were in reference to two or four proteins, respectively (putative geraniol 10-hydroxylase, hexosyltransferase or amine oxidase, pectin esterase 2, putative geraniol 10-hydroxylase, and putative LOV domain-containing, respectively). Subsequently, the best scored peptides were submitted to BIOPEP search (accessed on 15 February 2023, h http://www.uwm.edu.pl/biochemia/index.php/pl/biopep/) to hypothesize their possible bioactivities [29], but, so far, no bioactivity has been detected for these peptides.

Furthermore, nine and six peptides (before and after trypsinization, respectively) attributable to *Pseudomonas* sp. PIC25 resulted in a score higher than 0.6, but none of these were recognized in the proteins identified in Table 8; neither were they present in the BIOPEP database.

### 2.4. Untargeted Metabolomics of the Most Bioactive OP-F

Table 9 reports the percentages of different metabolites identified in the fifth OP1-F sample by GCMS analysis. Among these metabolites, a high percentage (about 25%) were represented by antioxidant compounds such as tyrosol and 4 hexylphenol, probably responsible for the highest antioxidant activity observed in the fifth peak (Table 6). Of interest was the 7.6% presence of glutamic acid, which is known to help in maintaining the integrity of the intestinal barrier, as it is incorporated into proteins during their synthesis by the “good” bacteria of the intestinal microbiota, thus, favoring their development [30,31].

### 2.5. Cellular Anti-Inflammatory Activities

To better clarify, at a molecular level, whether OP-W and OP-F aqueous extracts were able to modulate the expression of pro-inflammatory (IL-6 and TNF-α) and anti-inflammatory (IL-10) genes, a real time RT-PCR analysis was carried out on RNA of human PBMCs, which were previously in vitro supplemented, or not supplemented (CTRL), with the aqueous OP extracts (OP-W and OP-F; 2.5 μg/mL of extracts for 24 h) and then stimulated (s.) or not stimulated (n.s.) with LPS (100 ng/mL for 2.5 h). The mRNA expression levels of cytokines in OP-W s. and OP-F s. samples showed significant reductions for IL-6 (in both OP-W s. and OP-F s. cells; *p* < 0.05), IL-10 (*p* < 0.01 and < 0.05, respectively), and TNF-α (*p* < 0.01 and <0.05, respectively), when compared with not supplemented cells (CTRL s.) (Figure 2).

### 2.6. Cytokines Concentrations in Conditioned Medium

To confirm the anti-inflammatory effect of OP aqueous extracts (OP-W and OP-F) in LPS stimulated cells, the concentration of a panel of 16 cytokines (of IL-1α, IL-1β, IL-2, IL-4, IL-5, IL-6, IL-8, IL-10, IL-12, IL-13, IL-15, IL-17, IL-23, IFNγ, TNF-α and TNF-β) in PBMC culture supernatants was measured by a multiplex ELISA assay. Only the cytokines IL-6, IL-8 and TNF-α resulted as detectable (the remaining were below the detection levels) and, among these, both IL-6 and TNF-α resulted as significantly reduced by the OP-W extracts (*p* < 0.05) (Figure 3). Interestingly, the only cytokine that resulted as detectable also in the conditioned medium of cells not stimulated by LPS was IL-8 (means ± S.E.M. were 57.8 ± 13.1, 68.3 ± 16.2, 100.6 ± 54.5 pg/mL in CTRL n.s., OP-W n.s. and OP-F n.s. cells, respectively; *p* = n.s.).

## 3. Discussion

Our results regarding the phenolic composition of olive pomace extracts confirm that this by-product retains most bioactive compounds present in the fruit, despite the complex chemical transformations occurring during the fruit processing and pomace storage stages.

The chemical profile of this by-product has been widely described by several authors [32,33] and our results allow the conclusion that, despite a huge variability due to the impact of both endogenous (i.e., varieties) and exogenous (i.e., agro-pedoclimatic) factors [34,35], olive pomace represents a good source of natural health-promoting compounds in concentrations much higher than virgin olive oil [36,37]. In the present study too, the evaluation of three different olive pomace samples confirmed the presence of beneficial health compounds in all the batches investigated. These OP samples, although belonging to the same cultivar planted in the same territory, were different in terms of the period of olive harvest and, consequently, their ripening state.

The main phenolic compounds identified in the three pomace samples included hydroxytyrosol, tyrosol, oleuropein, ligstroside, pinoresinol, flavonoids, and hydroxycinnamic acids, and big differences were observed among samples [36]. OP1, the sample characterized by the highest content of unripened olives, was also the sample with the highest levels of target phenols, and with the highest antioxidant activity, based on the three in vitro tests used. Indeed, the ripening state strongly influences the phenolic content and antioxidant properties, that decrease along olive maturation [38].

This OP sample also conserved its highest antioxidant properties in both the aqueous absorbable and not absorbable digested extracts (OP-W n.a. and OP-W, Table 4), being the target phenols here mainly represented. Furthermore, the fifth fraction (OP-F; Table 5), obtained by the same OP1-W sample, also conserved a very high antioxidant property. This was probably due to the high content of tyrosol (20.49%) and 4 hexylphenol (4.52%), that together represented more than 25% of the total OP1-F sample, as testified by the metabolomic analysis (Table 8). Indeed, the free radical scavenging and metal-chelator properties of hydroxytyrosol and secoiridoid derivatives have been well recognized [39], and the high antioxidant efficiency has been attributed to the presence of the O-dihydroxymethyl moyety in the molecule, which mainly acts as a chain breaker by donating a hydrogen atom to peroxyl-radicals (ROO*). However, it has been proposed that hydroxytyrosol may confer an antioxidant protection by also reinforcing the endogenous defence systems against oxidative stress, by boosting different cellular signalling pathways [40]. A good antioxidant capacity has also been demonstrated for the 4-O-monohydroxy compounds ligstroside and tyrosol [41], and several interesting functional properties have been reported for oleuropein, including antioxidant, anti-inflammatory, anti-atherogenic, anti-cancer, and anti-microbial activities, among others [42]. Luteolin and apigenin have been confirmed as relevant components of olive pomace, as previously reported by Peralbo-Molina et al. [33], and are known to exert a protective effect against the deleterious effects of reactive oxygen [43].

However, in order to exert beneficial health effects, phenolic compounds in OP samples must be bioaccessible and bioavailable [44]. Bioaccessibility, which is defined as the release of a compound from its natural matrix to be available for intestinal absorption, is the first limiting factor for bioavailability. Ahmad-Qasen et al. [45] demonstrated that, after ingestion, the bioaccessibility of total phenolic compounds decreased during the first hour of the digestion process due to the degradation of the bioactive compounds following the pH variation and enzymatic activity, whereas for the rest of the digestion process, a constant value of TPC reached the duodenum. Furthermore, Ribeiro et al., 2020 [23], showed that after gastrointestinal digestion, more than 50% of the water-soluble compounds remained bioaccessible, especially hydroxytyrosol and potassium.

Bioavailability represents the ability of a nutrient or food bioactive to be efficiently digested, absorbed and distributed to provide its beneficial effect to the organism, participating in its physiological processes and storage [46].

Despite the huge research on the biological properties of olive phenolic compounds, information relative to their bioavailability, that strongly influences the pharmacological function, is limited. Recently, however, excellent reviews on the bioavailability of bisphenols and their metabolism have been proposed [46,47,48,49]. It has been shown that after oral administration, hydroxytyrosol is dose-dependently absorbed until the saturation of the phase I metabolic processes of intestinal transporters is reached. Thereafter, the absorption stops and hydroxytyrosol seems to undergo a rapid and intense metabolism [48,50], so that only a small fraction of free (unchanged) hydroxytyrosol is detectable in plasma. Indeed, the biological role of many bioactive compounds in the human organism is attributed to their metabolites, and the intestine represents the main site where it occurs for orally administered compounds [51]. In most human studies, the hydroxytyrosol absorption rate ranges from 55 to 90%, circulating bound to lipoproteins, reaching its maximal concentration after 1–2 h following its administration, then the molecule rapidly becomes undetectable. As proof of its absorption, the feces and urinary excretion of hydroxytyrosol and its metabolic derivatives have usually been adopted, with excretion rates resulting as maximum at 0–4 h [52,53].

Tyrosol has generally fewer studies due to its lower bioactivity, and also lower metabolites, according to the available information [46,51].

Our results showed that most phenolic compounds present in the olive pomace extracts were still detected at high levels in the absorbable fraction obtained after in vitro digestion, indicating in most cases a very high bioaccessibility. Hydroxytyrosol, for instance, showed bioaccessibility indexes higher than 80% in two out of three samples, and similar values were found for tyrosol in OP2. These results were in accordance with data reported by Ribeiro et al. [23], who obtained very similar values (82 and 77%, for hydroxytyrosol and tyrosol, respectively), demonstrating that they were the most bioaccessible compounds present in olive pomace. In a previous investigation, Seiquer et al. [54] also reported that the most bioaccessible and stable compounds after in vitro digestion of olive oil were tyrosol and hydroxytyrosol, and specifically, these compounds were absorbed in the intestine by passive diffusion as a result of their polar structure, thus, also demonstrating a good bioavailability, even though dissimilar to each other [47]. The passive diffusion of these compounds in the small intestine was recently confirmed by Sakavitsi et al. [51] who also observed a passive diffusion of caffeic acid, homovanillic acid, HT-3-*O*-sulphate, and 3,4-dihydroxyphenylacetic acid [48] as the main metabolites of hydroxytyrosol.

In OP1, the bioaccessibility index of tyrosol was higher than 100%, and the same was observed for caffeic acid. This was not surprising, suggesting that these molecules may be released from the food matrix and/or metabolized from other phenolic compounds with more complex structures [55].

Furthermore, bioavailability studies in the human body have demonstrated that absorbed oleuropein and tyrosol can be metabolized into free hydroxytyrosol, thus, increasing the concentration of HT in circulation [48,56].

However, potential beneficial effects have also been demonstrated for other compounds present in OP such as mineral, proteins and sugars [23]. In this study, the effects of sugars and minerals were not evaluated, while the possible presence of interesting peptides (after digestion of proteins) for their biological effects was investigated.

Mass spectrometry analysis of OP1-W peptide hydrolysates allowed the identification of 13 and 14 proteins specific to *Olea europaea* olive or to saprophytic microorganisms of the olive tree plant, respectively. The profile of these peptides was checked in the open-access PeptideRanker and BIOPEP tools. Overall, seven peptides of *Olea europaea* olive received a significant score. These peptides were referable to putative geraniol 10-hydroxylase, hexosyltransferase or amine oxidase, pectin esterase 2, putative geraniol 10-hydroxylase, and putative LOV domain-containing. However, none of the best scored peptides matched with the BIOPEP database.

Even if the presence of *Pseudomonas* sp. in OP samples could not be reconducted to any beneficial health effects for “consumers”, it was interesting to find traces of how plants and microorganisms join forces to address environmental pitfalls. Indeed, these strains of *Pseudomonas* spp. (PICF141 and PIC25) have shown high in vitro inhibition ability of pathogens’ growth such as *V. dahliae*, responsible for Verticillium wilt of olive [27]. Furthermore, fifteen peptides attributable to *Pseudomonas* sp. PIC25 showed a best score, but none were recognized in the proteins identified in Table 8, neither were they present in the BIOPEP database.

Finally, we tested, ex vivo, the effects of OP1-W and OP1-F extracts on cells present in the blood, the first tissue with which these extracts come into contact as soon as they are absorbed by the intestinal wall, even before they undergo phase I and II metabolism that favors their urinary excretion [57].

To date, no data exist on the effects of predigested phenols obtained from olive by-products, neither on human blood immune cells (PBMC) withdrawn from healthy individuals, or on in vitro cell cultures.

We found that the digested absorbable OP1-W extract showed significant anti-inflammatory activity on the ex vivo human PBMC stimulated by LPS. This finding was testified by the significant reduction in pro-inflammatory cytokines IL-6 and TNF-α in conditioned medium (*p* < 0.05; Figure 3) and by a lowering trend of IL-8 together with a significantly lower expression of mRNAs encoding IL-6 and TNF-α (*p* < 0.5 and *p* < 0.01, respectively; Figure 2).

Furthermore, in the ex vivo evaluations, a significant down-regulation of IL-10, a typical anti-inflammatory gene, was also observed. Even if this finding seems in contrast with the previous results, we assumed that it was due to a reduced triggering of the inflammatory process (and its subsequent cascade of pro- and anti-inflammatory signals) in PBMC pre-treated with OP-W (for 22 h) in response to a pro-inflammatory stimulus (LPS), rather than to a lower ability to induce the IL-10 gene to turn off an inflammation fully triggered. This finding was in accordance with results reported by Camargo et al. [14], who observed that the dietary assumption of high-phenol virgin olive oil in patients suffering metabolic syndrome, by switching the activity of peripheral blood mononuclear cells to a less deleterious inflammatory profile, was able to repress the inflammatory process, even if it occurred because of the addition of a stimulus of inflammation (i.e., LPS).

Indeed, LPS normally acts by activating the nuclear factor-kappa B (NF-κB) and mitogen-activated protein kinase (MAPK) pathways, causing the overexpression of various inflammatory mediators, such TNF-α, IL-1β, IL-6, nitric oxide (NO) and prostaglandin E2 (PGE2) [58,59]. However, Camargo et al. [14] found a chemokine repression as a direct consequence of phenols interaction with NF-κB/MAPK/AP-1 inflammation signaling pathways.

Recent research has found evidence of the ability of the phenols of several plants to induce cellular, biochemical, and epigenetic modifications, resulting in modulation of the homeostasis of key cellular processes such as the control of oxidative stress, inflammatory response, and gene expression, among others [57]. Wang et al. [60] recently showed that pretreatment with tyrosol markedly inhibited the activation of NF-κB and apolipoprotein-1 (AP-1) in LPS-induced A549 cells.

In the present study, the fifth absorbable OP1-F aqueous extract, which was constituted of more than 20% of tyrosol and utilized at the same concentration of OP-W, significantly reduced the expression of mRNAs encoding IL-6, IL-10, and TNF-α (*p* < 0.05); however, it was not able to significantly reduce the cytokine levels in the conditioned medium. Indeed, an increasing trend was observed, although not statistically significant, for all three detectable cytokines. We supposed that the lack of ability of the OP1-F extract (used at the same concentration of OP-W) vs. OP1-W sample to reduce the concentration of cytokines IL-8 and TNF-α was due to a too-high concentration of tyrosol (by using the OP1-F extract at the same concentration of OP1-W extract, the tyrosol level resulted as much higher) or to the presence, in the OP1-W extract, of other compounds eliciting a synergistic effect with tyrosol in quenching the inflammatory response elicited by LPS stimulus.

Indeed, we supposed that a too-high level of tyrosol could induce a sort of paradoxical effect [61,62] as seems stated by the higher IL-8 level in PBMC supplemented with OP1-F aqueous extract, but not triggered by LPS. Indeed, in the present study, as observed by other authors [24], in Caco-2 cell culture maintained at basal condition (control), the only interleukin detectable in the basal medium was IL-8. However, while in the presence of the extract OP1-W extract, the interleukin IL-8 showed a decreasing trend; in the presence of OP1-F extract, the trend was the opposite.

Various reports have stated that diets with high contents of polyphenols are associated with a reduced production of these inflammatory cytokines and a consequent improvement of inflammation [10]. In the present study, for the first time, similar results exerted by digested phenols derived from olive pomace on the ex vivo cultured human (healthy) PBMCs have been demonstrated.

Interestingly, the concentration of phenols in the dose of OP-W extract here adopted on cells was in the order of nanogram. Even if it was quite difficult to find and compare data relative to the bioavailability of oil phenols or phenols of olive by-products orally administered in human in vivo studies [46,47,53], in blood, what was possible to infer from the literature was that it is not impossible to reach this nanogram concentration, although for a very short time, by ingesting, for example, a quantity corresponding to 10 g of olives. Therefore, these preliminary results relative to the anti-inflammatory properties exerted by the mixture of phenols present in the in vitro absorbable digested olive pomace seem promising and worthy of further investigation.

An expanding body of literature has shown that, through increasingly eco-friendly and cost-effective extraction processes applied to olive oil processing by-products [63,64], it is possible to obtain high quantities of bioactive compounds which have been proven to be effective in exerting beneficial effects on health. Our data further demonstrate that olive pomace can, therefore, become a valuable raw material that can be used not only in the energy sectors, food, cosmetic and animal feed [7,65], but also in the nutraceutical field.

## 4. Materials and Methods

### 4.1. Chemicals and Reagents

All reagents were of analytical purity. The 2,2-diphenyl-1-picrylhydrazyl (DPPH), ABTS diammonium salt (2,2-azinobis-3-ethylbenzothiazoline-6-sulphonic acid), Folin–Ciocalteu’s reagent, standards of Trolox and gallic acid were purchased from Sigma-Aldrich Corp. (Milan, Italy). Bile salts were from Oxoid™ (Hampshire, U.K.). Spectrapor 3 45 MM/15 M membrane (cut-off 3.5 kDa) was purchased from Fischer Scientific (Milan, Italy). Pancreatin (from porcine pancreas: P3292-100G), pepsin (from pig gastric mucosa: ≈2500 units/mg protein), α-amylase from human saliva (A0521-500 units/mg), formic acid, potassium sorbate, sodium carbonate, trifluoroacetic acid (TFA), and peptidyl-dipeptidase were purchased from Sigma-Aldrich Corp. (Milan, Italy).

Hydroxytyrosol, tyrosol, oleuropeion, apigenin, luteolin and myricetin were purchased from Extrasynthese (Lyon, France); caffeic acid, chlorogenic acid, pinoresinol, and ligstroside were purchased from Sigma-Aldrich Corp. (Milan, Italy). Acetonitrile was purchased from Pai Acs, Panreac. All other chemicals and solvents were of the highest analytical grade from Sigma-Aldrich Co. (St. Louis, MO, USA).

### 4.2. Olive Pomace Methanolic Extracts

Three batches of sun-dried OP were gently provided by Oil Mill Industry Consoli (Adrano, Catania, Italy) and transported to the laboratory where they were packed in polyethene bags and kept in a freezer at −80 °C until analysis. OP1 was collected in September, OP2 in October, and OP3 in November/December, thus, the ripening state increased with each harvest of olives. These OP samples were stratified (and stabilized with Consoli’s patent), one batch on top of the other, in a unique big pool, and stabilized with Consoli’s patent (Consoli’s patent 0001428707). The pool was well covered until the following summer season, when the pool was opened, and the OP was dried under the sun. Olive pomace samples were collected from the bottom of the pool at the end of June (OP1), July (OP2), and August (OP3). These OP samples were composed mainly of the olive cultivar *Olea europaea L*. (*Nocellara etnea* as main cultivar) and the proximate composition as reported in Chiofalo et al. [66].

From each OP sample (OP1, OP2 and OP3; 5 g/batch), methanolic extracts were obtained by using a Soxhlet apparatus for 5 h to fall and 200 mL of pure methanol. The obtained solutions were then evaporated with a Rotavapor (Buchi B-490) to collect the OP extracts for further characterization. The crude extracts (methanolic OP extracts) were then weighed (1.228 g, 0.967 g, and 1.004 g, respectively) and the yield calculated (24.6%, 19.34%, and 20.1%, respectively).

#### 4.2.1. Radical Scavenging Activity Assays in Methanolic OP Extracts

By using the 1,1-diphenyl-2-picrylhydrazyl (DPPH) method, the radical scavenging activity of the methanolic OP1, OP2 and OP3 extracts was measured [67]. In a 96-multiwell plate, 50 μL aliquot of each OP extract (0–2 mg/mL) or of the standard Trolox (0–100 mg/mL), in triplicate, was added to 200 μL of DPPH solution (0.1 mM in methanol). After incubation in darkness for 30 min at 37 °C, the absorbance was measured at 490 nm using a UV–VIS microplate reader (FLUOstar Optima, BMG Labtech, Ortenberg, Germany) against DPPH solution as a blank. Values were expressed as Trolox equivalent (μg TE/mg dry extract).

The radical cation scavenging activity of each extract was measured using the 2-2′- azino-bis (3-ethylbenzo-thiazoline-6-sulphonate) diammonium salt (ABTS) method [68]. In a 96-multiwell plate, 50 μL aliquot of sample (0–5 mg/mL) was added to 200 μL of ABTS solution (5 mM). ATBS solution was derived by oxidizing ABTS with MnO_2_ in distilled water for 30 min in the dark, and then the solution was filtered through filter paper. After 20 min incubation in darkness at room temperature, the absorbance was read at 734 nm using a UV–VIS microplate reader (FLUOstar Optima, BMG Labtech, Ortenberg, Germany) against ABTS solution as a blank. Values were expressed as Trolox equivalent (μg TE/mg dry extract).

#### 4.2.2. Ferric Reducing Antioxidant Power (FRAP) Assay in Methanolic OP Extracts

The reducing power of methanolic OP1, OP2 and OP3 extracts was evaluated according to a ferric reducing antioxidant power (FRAP) assay [69]. In a 96-multiwell plate, 25 μL aliquot of sample (0–2 mg/mL) or of standard Trolox (0–100 μg/mL) was added to 175 μL of FRAP working solution containing 20 mmol/L ferric chloride, 300 mmol/L acetate buffer (pH 3.6), and 10 mmol/L TPTZ (2,4,6- tri (2-pyridyl)—S-triazine) made up in 40 mmol/L HCl. The three solutions were mixed at a 10:1:1 ratio (*v*:*v*:*v*). The mixture was incubated in darkness for 30 min at 37 °C and then the absorbance was determined using a UV–VIS microplate reader (FLUOstar Optima, BMG Labtech, Ortenberg, Germany) at 593 against FRAP solution as a blank. Values were expressed as Trolox equivalent (μg TE/mg dry extract).

#### 4.2.3. TPC of Methanolic OP Extracts

The Folin–Ciocalteu method was used to determine TPC [70]. Briefly, 25 μL aliquots of OP1, OP2 and OP3 extracts (5 mg/mL) were incubated for 5 min with 125 μL of 10% (*w*/*v*) Folin–Ciocalteu reagent. After the addition of 125 μL of Na_2_CO_3_ (10% *w*/*v*) and incubation for 30 min in darkness at room temperature, the absorbance was read using a UV–VIS microplate reader (FLUOstar Optima, BMG Labtech, Ortenberg, Germany) at 320 nm. The results were derived from a gallic acid calibration curve (0–1000 ug/mL) prepared from a stock solution (1 mg/mL in ethanol). Values were expressed as mg of gallic acid equivalents (GAE) per gram of dried weight extract (mg of GAE/g extract).

### 4.3. OP In Vitro Digestion and OP Aqueous Extracts

To mimic the in vitro oral, gastric and intestinal digestion of OP samples, the procedure indicated by Diab et al. [26] was followed. Briefly, 5 g of each OP batch were mixed with 25 mL SSF and 3 mL (stock 75 U/mL) α-salivary amylase (from human saliva), 5.8 mL distilled H_2_O, and 0.2 mL CaCl_2_, and then incubated on a magnetic stirrer for 2 min at 37 °C (oral digestion). For the gastric digestion, 40 mL SGF, 7 mL pepsin (stock 25,000 U/mL), 3 mL distilled H_2_O, and 0.03 mL CaCl_2_ were added to the oral outcome and the pH was lowered to 3.0 by HCl; the mixture was incubated for 2 h at 37 °C on a magnetic stirrer, and the pH was checked regularly. Finally, 50 mL of gastric outcome was mixed with 20 mL of pancreatin (stock 100 U/mL), 50 mL of SIF, 6 mL distilled H_2_O, 10 mL bile salt (stock 10 mM), 0.024 mL CaCl_2_, and 0.7 mL of 1 M HCl to neutralize the pH to 7.0 to simulate the intestinal digestion.

At the end of this process, the obtained mixture was incubated on a magnetic stirrer for 2 h at 37 °C. Then, to inactivate the enzymes used in the digestion process, the mixture was heated to 90 °C for 10 min. Eventually, the mixture was dialyzed with membrane cut-off 3.5 kDa (Spectra/Por molecular porous membrane tubing, Thermo Fisher Scientific, Milan, Italy) against 250 mL of water for 24 h at 4 °C to separate the high molecular weight (mw) fraction (>3.5 kDa, inside the dialysis membrane) from the low mw fraction (<3.5 kDa, outside the dialysis membrane). At the end of the incubation process, the solution outside the dialysis tubing (OP-W) represented the aqueous OP sample that was available for absorption (whole serum-available sample) and the solution that had not managed to diffuse through the dialysis tubing (OP-W n.a.) represented the whole non-absorbable sample (colon-available) [23]. The dialyzed digested OP-W and OP-W n.a. extracts were then lyophilized, obtaining the OP digested aqueous extracts that were further characterized for their bioactive properties.

#### 4.3.1. Radical Scavenging Activity Assays in Aqueous OP-W n.a. and OP-W Extracts

The ABTS and DPPH assays described at Section 4.2.1. were utilized to determine the radical scavenging activities of the three aqueous OP-W n.a. and OP-W (5 mg/mL) extracts.

#### 4.3.2. Ferric Reducing Antioxidant Power (FRAP) Assay in Aqueous OP-W n.a. and OP-W Extracts

The FRAP assay described at Section 4.2.2. was utilized to determine the ferric reducing antioxidant powers of the OP-W n.a. and OP-W (5 mg/mL).

#### 4.3.3. TPC in OP-W n.a., OP-W

The TPC assay described at Section 4.2.3. was utilized to evaluate the total phenolic contents in the OP-W n.a. and OP-W samples.

#### 4.3.4. HPLC-DAD Analysis of Methanolic and Aqueous OP Extracts

The analysis of phenolic compounds in methanolic and aqueous OP samples were carried out with a Jasco (Tokyo, Japan) HPLC-DAD system, consisting of a PU-4180 pump, a MD-4015 PDA detector, and an AS-4050 autosampler. An Agilent Zorbax Eclipse Plus C18 reverse-phase column (Agilent, Santa Clara, CA, USA.) (4.6 × 100 mm I.D, 3.5 μm) was used as a stationary phase. Two different methods, modified from that of Peršurić et al. [71], were applied for the separation and identification of the analytes in the mixture, both employing water adjusted to pH 2.5 with orto-phosphoric acid (Solvent A) and acetonitrile (Solvent B) as the mobile phase. For the identification and quantification of hydroxytyrosol, tyrosol, oleuropein, apigenin, luteolin, myricetin, ligstroside, and pinoresinol, the following step gradient was used: 90 to 72% A for 10 min, 72% for 15 min, 72 for 70% for 10 min, 70% for 10 min, 70 to 5% for 10 min, and kept constant for 5 min. The gradient was restored to initial conditions and kept constant for 20 min for re-equilibration. The flow rate was 0.5 mL/min, the injection volume for both reference standards and samples was 50 μL, and the detection wavelengths were set at 280 and 360 nm.

For the identification and quantification of phenolic acids (gallic, caffeic, and chlorogenic acid) a different step gradient was used: 97% A for 6 min, 97 to 85% for 11 min, 85 to 82.8% for 10 min, 82.8 to 50% for 13 min, back to initial conditions for 3 min, and constant for 13 min for re-equilibration. The injection volume was 20 μL, the flow rate was 0.7 mL/min, and the PDA wavelengths were set at 280, 329 and 360 nm.

To quantify the analytes, calibration curves were constructed for each standard by injecting six different concentrations (50 ppm, 25 ppm, 12.5 ppm, 6.25 ppm, 3.12 ppm and 1.56 ppm) in duplicate. Stock solutions of hydroxytyrosol, tyrosol, oleuropein, apigenin, luteolin, myricetin, ligstroside, and pinoresinol were prepared with DMSO/acetonitrile (20/80) mixture at a concentration of 2 mg/mL. The dilutions were then carried out with water adjusted to pH 2.5 with orto-phosphoric acid to ensure greater stability of the analytes. Gallic acid was dissolved in water, while caffeic and chlorogenic acid were dissolved in methanol and further diluted with the mobile phase.

The bioaccessibility index was calculated as the percentage of the compound detected in the digested and lyophilized samples, with reference to the undigested sample.

#### 4.3.5. HPLC-DAD Fractioning of OP-W (OP-F)

The most bioactive OP-W extract (about 8 mg), based on its antioxidant properties, was further purified using reversed-phase high-performance chromatography on a 10-mm × 250 mm semipreparative C18 column (Supelco, Bellefonte, PA, U.S.A.), equilibrated in water and eluted at a flow rate of 2 mL/min with a discontinuous gradient of acetonitrile. In this way, seven OP-W fractions (OP-F) were obtained, and the ABTS and TCP were determined on those OP-Fs.

#### 4.3.6. LC-MS/MS Peptide Profiling of OP-W

The most bioactive OP-W extract was resuspended in 50 mM ammonium bicarbonate, pH 8.0, reduced with 10 mM DTT at 56 °C for 45 min, and alkylated with a 55 mM solution of iodoacetamide for 30 min at room temperature in the dark and then desalted by a SEP-PAK chromatography. The main fraction was manually collected and lyophilized. An aliquot of the sample was directly analyzed by LC-MS/MS for protein identification. The remaining portion of the lyophilized fraction was resuspended in 50 mM ammonium bicarbonate, pH 8.0, and incubated with trypsin in a 1/50 ratio (*w*/*w*) at 37 °C for 2 h. The sample was acidified with a final concentration of 0.2% trifluoracetic acid. The peptide mixture was first concentrated and desalted by C18 zip-tip and then was lyophilized. The lyophilized fraction was resuspended in 0.2% HCOOH and analyzed by LC-MS/MS, using a 6530 Q- TOF LC/MS (Agilent) system equipped with a nano-HPLC. After loading, peptide mixtures were first concentrated and desalted on the pre-column. For protein identification, the raw data obtained from the LC-MS/MS analysis were used to search both “Olea” and “Olea Europea” protein databases by an in-house version of the Mascot software.

#### 4.3.7. Search of Potential Biological Activities and Peptide Ranking

The potential bioactivities of OP-W peptides were predicted using the open access tool PeptideRanker (accessed on 15 February 2023, http://bioware.ucd.ie/compass/biowareweb/) [72], a web-based tool used to predict the probability of biological activity of peptide sequences. PeptideRanker provides peptide scores in the range of 0–1. The maximum scores indicate the most active peptides, whereas the minimum scores denote the least active peptides. Here, only those peptides with a score higher than 0.6 were considered as potentially “bioactive”. Subsequently, the lists of best-ranked peptides were submitted to the web-available database BIOPEP (accessed on 15 February 2023, http://www.uwm.edu.pl/biochemia/index.php/pl/biopep/) which contains collected data relative to peptides with a recognized bioactivity.

#### 4.3.8. Untargeted Metabolomics of OP-F by GCMS Analysis

Due to the small quantity of each of the seven separated OP-F, only the most promising fraction (the fifth fraction), based on ABTS and TPC results, was referred to untargeted metabolomics by GCMS Analysis. GC/MS analysis was performed by a 7820A (Agilent Technologies, Santa Clara, CA, USA) with a HB-5 ms capillary column (30 m × 0.25 mm × 0.25 µm film thickness) (Agilent Technologies). The injector, ion source, quadrupole, and GC/MS interface temperature were 230, 230, 150, and 280 °C, respectively. The flow rate of helium carrier gas was kept at 1 mL/min. An amount of 1 µL of derivatized sample was injected with a 3 min solvent delay time and split ratio of 10:1. The initial column temperature was 40 °C and held for 2 min, ramped to 150 °C at a rate of 15 °C/min, and held 1 min, and then finally increased to 280 °C, at a rate of 30 °C/min, and kept at this temperature for 5 min. The ionization was carried out in the electron impact (EI) mode at 70 eV. The MS data were acquired in full scan mode from m/z 40–400 with an acquisition frequency of 12.8 scans per second. Compound identification was confirmed by injection of pure standards and comparison of the retention time and corresponding EI MS spectra.

### 4.4. Human PBMC Culture, Supplementation and RNA Extraction

Human peripheral blood mononuclear cells were isolated from fresh heparinized blood samples (20 mL) obtained from five healthy donors. Cells were separated by gradient centrifugation and then the number of live mononuclear cells, suspended in complete medium containing RPMI 1640 Medium (Thermo Fischer Scientific, Waltham, MA, USA) with 10% heat inactivated fetal bovine serum (Gibco^TM^, Thermo Fischer Scientific, Waltham, MA, U.S.A.), 100 μg/mL streptomycin (Biochrom^AG^, Berlin, Germany), 2 mM L-glutamine (Euroclone^®^, Milan, Italy) and 100 units/mL penicillin (Biochrom^AG^, Berlin, Germany), was determined using a counting chamber and the Trypan blue dye exclusion procedure [70]. The final number of live cells was adjusted to 4 × 10^6^/well (2 mL) and blood cells of each donor were cultured in triplicate at 37 °C in 5% CO_2_, supplemented or not (CTRL) with OP-W and OP-F aqueous extracts (final concentration of 2.5 μg/mL) that were previously filtered on 0.22 μm acetate cellulose filters. The aqueous extract concentrations were chosen after preliminary tests and, based on the yield obtained after dialysis (8.9%), the OP-W extract contained 0.87, 1.4, 0.3, 0.36, and 0.38 μg/mg of hydroxytyrosol, tyrosol, oleuropein, myricetin, and luteolin, respectively.

After 24 h of incubation, one half of cells/donor (CTRL, OP-W and OP-F cells/donor) was triggered (stimulated: s.) or not (n.s.) with LPS (100 ng/mL) for 2.5 h. Later, cells were transferred in Eppendorf vials (2 mL) and centrifugated (× 300 g) to separate the conditioned media for cytokine measurements. Finally, the residual cells were washed twice with phosphate-buffered saline (Gibco^TM^, Thermo Fischer Scientific, Waltham, MA, USA) and finally stored at −80 °C until RNA extraction by Mini Kit (QIAGEN GmbH, Hilden, Germany).

A NanoVue Spectrophotometer (GE Healthcare, Milano, Italy) was used to measure RNA yield and purity. Only samples with A260/A280 ratio > 1.8 were used.

### 4.5. Analysis of mRNA Levels by Real Time Reverse Transcriptase-Polymerase Chain Reaction (Real Time RT-PCR)

To obtain cDNA, 1 μg of RNA for each sample was reverse transcribed using an iScript cDNA Synthesis Kit (Bio-Rad Laboratories, Hercules, CA, U.S.A.). The subsequent PCR was performed in a total volume of 10 μL containing 2.5 μL (12.5 ng) of cDNA, 2 μL of RNAsi free dH_2_O, 5 μL SsoAdvanced Universal SYBR Green Supermix (Bio-Rad Laboratories), and 0.5 μL (500 nM) of each primer. The investigated genes were IL-6, IL-10, and TNFα. All primers (listed in Table 10) were purchased from Sigma-Aldrich Life Science Co. LLC. (USA) and were intended for human cells. The 18S gene was used as the reference gene.

### 4.6. Measurment of PBMC Cytokines in Conditioned Medium

The pro- and anti-inflammatory cytokines IL-1α, IL-1β, IL-2, IL-4, IL -5, IL-6, IL-8, IL-10, IL-12, IL-13, IL-15, IL-17, IL-23, IFNγ, TNF-α and TNF-β were estimated in the culture conditioned medium of human PBMC, by using multiplex immunoassay (Q-Plex Human Cytokine—Screen 16-plex, Quansys Biosciences, Technogenetics Srl., Milan, Italy), Q-View Imager LS, Q-View software, and following the manufacturer’s instructions. The culture medium was obtained from PBMC previously cultured or not with OP-W and OP-F (2.5 μg/mL for 24 h) and then stimulated (s.) or not (n.s.) by LPS (100 ng/mL for 2.5 h).

### 4.7. Statistical Analysis

Data relative to real time RT-PCR and cytokine concentrations were analyzed by using the two-tailed paired *t*-test, whereas one-way ANOVA with Tukey’s multiple comparison test was adopted for data relative to radical scavenging activities and phenolic characterization (Prism 7, GraphPad Software Inc., San Diego, U.S.A.) considering significant differences for *p* < 0.05. Values were expressed as mean (S.D.) unless otherwise stated.

## 5. Conclusions

In recent years, there has been growing interest in natural substances as possible sources of active compounds for disease prevention and/or health benefits. The greater awareness of the need to reduce environmental impact and to better exploit resources still available, has led researchers to focus their efforts on the possibility of identifying beneficial molecules for the organism from by-products of the food chain. In this context of a circular economy, the olive processing cycle could provide an example of reuse of waste products. The pomace and vegetative waters are, in fact, rich with very interesting molecules, among which are luteolin, with known anti-tumoral properties; hydroxytyrosol, having hypocholesterolemic action; and tyrosyl, ligstroside, and oleuropein.

In the present research, the phenolic composition and antioxidant activities of three batches of olive pomaces, differing by ripening state, and evaluated both before and after an in vitro simulation of the digestive process, revealed very different contents of phenols (hydroxytyrosol, tyrosol, oleuropein, ligstroside, pinoresinol, flavonoids, and hydroxycinnamic acids) and antioxidant properties, thus, confirming the great influence of the ripening stage and storage conditions on phenolic composition.

Although only part of ingested phenolics can pass the gut barrier, it was noteworthy that these nutraceutical compounds were still present in the digested absorbable aqueous extract of olive pomace and were able to exert an interesting anti-inflammatory activity both at transcriptional level and in the surnatants of human ex vivo PBMC cultures. The data here presented were in line with the in vivo repression of several pro-inflammatory genes observed in the only other work found in the literature that used PBMC cells from patients with metabolic syndrome [14].

However, further studies are necessary to determine the cause of the lower anti-inflammatory response obtained with OP-F, compared with the OP-W sample. Indeed, it could be connected to a higher bioactivity orchestrated by the entire phyto complex (synergistic effect) present in the OP-W sample, probably able to involve different pathways/mechanisms, or conversely, to an excessively high dose of tyrosol in OP-F. Nature and/or evolution could have determined the fast ability to metabolize and eliminate the excessive ingestion of phenols just to reduce deleterious effects when too concentrated in circulation.

The in vitro experiments represent a consistent approach to evaluating the health effect of new functional ingredients, but despite the inherent “défaillance” in in vitro simulation of digestion which is a rather complex physiological process, we believe that compounds under in vitro testing should always undergo a preventive in vitro digestion before their evaluation in cell cultures. Undoubtedly, the in vitro digestion represents one of the experimental approaches to encompass the open questions of bioavailability and metabolism food bioactives, particularly of phenols.

In the present research, however, it was not considered whether, and to what extent, the microbiota could affect digestion and bioavailability of the not absorbable olive pomace extract which, therefore, could also contribute to the increased bioavailability of bioactive compounds.

Furthermore, in the digested and absorbable sample used for cell testing, potentially bioactive peptides were also identified. Those resulting peptides have not yet been described, and they will be further investigated together with the characterization of polysaccharides and minerals, which are also potentially bioactive.

In the future, we would like to evaluate the potential immunomodulatory activity of these aqueous extracts, and, for this purpose, we believe that PBMCs taken from patients with autoimmune diseases could represent a consistent ex vivo model that could provide interesting outcomes.

The results herein reported clearly evidence the anti-inflammatory effect of digested aqueous OP extract and pave the way to an exploitation of the olive pomace by-product as a functional ingredient or as a nutraceutical.

## Figures and Tables

**Figure 1 molecules-28-02122-f001:**
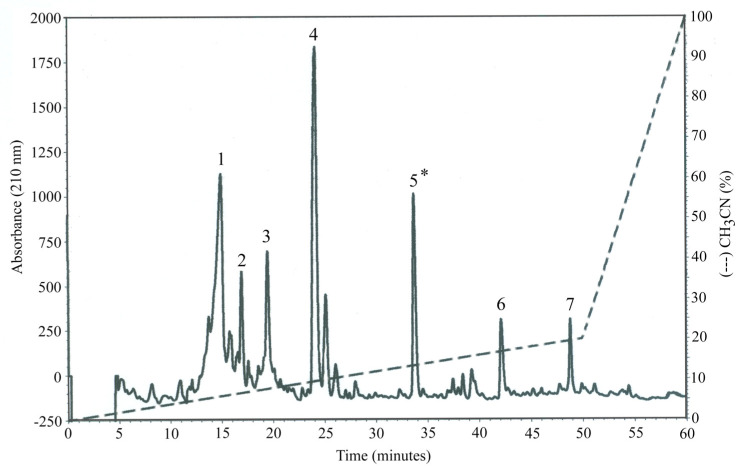
Major chromatographic peaks (n.7) obtained from OP-W1. The most bioactive aqueous adsorbable extract result was the fifth one (*), by HPLC-DAD.

**Figure 2 molecules-28-02122-f002:**
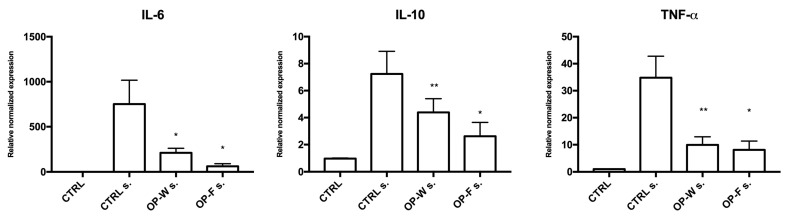
Real time PCR relative quantification of mRNAs encoding IL-6, IL-10, and TNF-α (± S.E.M.), evaluated in human PBMCs after stimulation (s) or not (n.s.) with LPS for 2 h in an incubator at 37 °C, 5% CO_2_. Blood cells were previously cultured in presence or not (CTRL) of aqueous extracts obtained from OP (OP-W and OP-F) after in vitro digestion and dialysis. * *p* < 0.05, ** *p* < 0.01.

**Figure 3 molecules-28-02122-f003:**
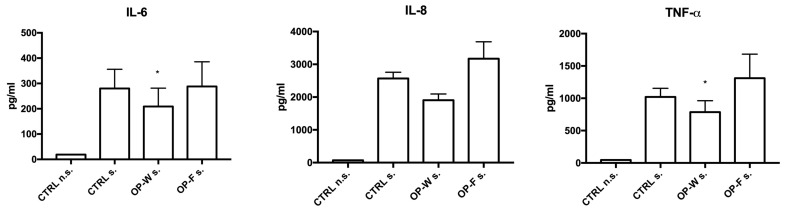
Cytokine concentrations evaluated in the culture medium of human PBMCs after stimulation (s), or not (n.s.), with LPS for 2 h in an incubator at 37 °C, 5% CO_2_. Blood cells were previously cultured in presence, or not (CTRL), of aqueous extracts obtained from OP after in vitro digestion and dialysis. * *p* < 0.05.

**Table 1 molecules-28-02122-t001:** Phenolic composition (µg/g dry weight) and TPC (mg of GAE/g extract) of the three OP batches.

Sample	Ht	T	Ole	Lig	Pin	Myr	Lut	Api	CA	ChlA	GA	TTMI	TPC
OP1	224.6 ± 0.4 ^a^	222.9 ± 1.8 ^a^	128.9 ± 1.9 ^a^	103.7 ± 0.5 ^a^	36.9 ± 0.9 ^a^	91.9 ± 0.5 ^a^	599.9 ± 2.7 ^a^	221.5 ± 5.5 ^a^	7.8 ± 1.1 ^a^	48.4 ± 4.3 ^a^	nd	1686.5 ^a^	99.8 ± 8.5 ^A^
OP2	36.9 ± 0.5 ^b^	46.7 ± 2.5 ^b^	44.9 ± 1.7 ^b^	27.1 ± 0.1 ^b^	24.0 ± 1.3 ^b^	70.8 ± 1.2 ^b^	474.1 ± 1.7 ^b^	175.5 ± 0.2 ^b^	9.8 ± 0.1 ^b^	28.9 ± 3.2 ^b^	nd	938.7 ^b^	26.3 ± 3.9 ^B^
OP3	4.9 ± 0.3 ^c^	8.2 ± 0.5 ^c^	29.4 ± 0.1 ^c^	24.1 ± 0.6 ^c^	17.8 ± 0.7 ^c^	72.9 ± 1.1 ^b^	354.7 ± 3.2 ^c^	129.9 ± 1.1 ^c^	7.5 ± 0.5 ^a^	11.4 ± 1.5 ^c^	nd	660.8 ^c^	14.5 ± 2.4 ^C^

Ht = hydroxytyrosol, T = tyrosol, Ole = oleuropein, Lig = ligstroside, Pin = pinoresinol, Myr = myricetin, Lut = luteolin, Api = apigenin, CA = caffeic acid, ChlA = chlorogenic acid, GA = gallic acid, TTMI = total targeted metabolite index, TPC = total phenolic content, nd = not detected. Superscript letters ^A,B,C,a,b,c^ within the same column refer to statistical analysis, and different letters indicate significant differences for *p* < 0.0001 and 0.5, respectively.

**Table 2 molecules-28-02122-t002:** Phenolic composition (total µg in the dialyzed samples) and TPC (mg of GAE/g extract) of the digested OP batches. See Table 1 for abbreviations.

Sample	Ht	T	Ole	Lig	Pin	Myr	Lut	Api	CA	ChlA	GA	TTMI	TPC
OP1-W n.a.	138.8 ± 0.4 ^A^	241.6 ± 6.7 ^A^	41.7 ± 3.0 ^A^	28.2 ± 2.2	50.6 ± 0.8 ^A^	201.0 ± 0.1 ^A^	212.2 ± 1.9 ^A^	nd	30.5 ± 2.8 ^A^	32.7 ± 1.3 ^A^	76.2 ± 1.4 ^A^	1053.5	127.9 ± 4.1 ^A^
OP2-W n.a.	59.4 ± 1.1 ^B^	101.0 ± 4.5 ^B^	37.8 ± 1.2 ^B^	nd	46.8 ± 3.4 ^B^	188.7 ± 2.1 ^B^	197.5 ± 1.6 ^B^	nd	24.8 ± 0.6 ^B^	28.9 ± 3.0 ^A^	93.8 ± 1.3 ^B^	778.7	96.8 ± 10.6 ^B^
OP3-W n.a.	nd	nd	12.8 ± 1.0 ^C^	nd	48.1 ± 1.3 ^AB^	183.0 ± 2.3 ^B^	192.2 ± 1.0 ^B^	nd	18.0 ± 1.2 ^C^	18.4 ± 1.1 ^B^	295.4 ± 2.1 ^C^	767.9	88.5 ± 21.7 ^B^
OP1-W	386.7 ± 3.1 ^a^	627.8 ± 3.4 ^a^	129.8 ± 4.8 ^a^	23.9 ± 1.3 ^a^	41.4 ± 1.7 ^a^	161.6 ± 1.5 ^a^	168.4 ± 1.0 ^a^	nd	36.2 ± 0.5 ^a^	57.5 ± 2.1 ^a^	56.0 ± 0.6 ^a^	1527.7	96.1 ± 3.0 ^a^
OP2-W	62.5 ± 1.4 ^b^	77.7 ± 1.8 ^b^	13.1 ± 1.1 ^b^	15.4 ± 0.9 ^b^	28.7 ± 0.8 ^b^	118.9 ± 0.3 ^b^	124.4 ± 0.7 ^b^	nd	14.1 ± 0.8 ^b^	26.9 ± 1.4 ^b^	34.9 ± 0.7 ^b^	516.6	72.4 ± 13.9 ^b^
OP3-W	nd	nd	23.9 ± 1.1 ^c^	20.7 ± 1.7 ^a^	26.3 ± 1.8 ^b^	100.5 ± 0.3 ^c^	105.1 ± 0.1 ^c^	nd	12.8 ± 1.7 ^b^	15.0 ± 1.3 ^c^	34.1 ± 3.1 ^b^	338.4	68.5 ± 7.6 ^b^

Superscript letters within the same column refer to statistical analysis. Uppercase letters ^A,B,C^ refer to non-absorbable aqueous digested samples (OP-W n.a.); lowercase letters ^a,b,c^ refer to absorbable aqueous digested samples (OP-W). Different letters indicate significant differences for *p* < 0.05.

**Table 3 molecules-28-02122-t003:** Bioaccessibility index (%) of metabolites in the absorbable aqueous-digested OP samples (OP-W). See Table 1 for abbreviations.

Sample	Ht	T	Ole	Lig	Pin	Myr	Lut	Api	CA	ChlA	GA
OP1-W	86.1	140.8	50.3	11.5	56.1	87.9	14.0	-	231.2	59.4	-
OP2-W	84.5	83.2	14.6	28.3	60.0	84.0	13.1	-	71.5	46.6	-
OP3-W	-	-	40.7	43.0	74.0	68.9	14.8	-	84.9	66.1	-

**Table 4 molecules-28-02122-t004:** The radical scavenging activity of methanolic extracts obtained from the three crude OP1, OP2 and OP3 batches. Values are reported as Trolox equivalent (μg TE/mg dry extract).

Radical Scavenging Assays			
	ABTS	DPPH	FRAP
Samples	Trolox Equivalent ± SD	Trolox Equivalent ± SD	Trolox Equivalent ± SD
OP1	124.6 ± 4.2 ^a^	44.5 ± 2.5 ^a^	74.6 ± 3.5 ^a^
OP2	55.6 ± 4.1 ^b^	19.6 ± 2.4 ^b^	31.5 ± 1.5 ^b^
OP3	19.5 ± 1.2 ^c^	9.22 ± 0.7 ^c^	6.3 ± 0.07 ^c^

^a,b,c^ different letters mean significant differences; *p* < 0.05.

**Table 5 molecules-28-02122-t005:** The radical scavenging activity of aqueous OP extracts after in vitro digestion and dialysis. Values are reported as Trolox equivalent (μg TE/mg dry extract).

	ABTS	FRAP
Samples	Trolox Equivalent ± SD	Trolox Equivalent ± SD
OP-W n.a. ^1^		
1	127.3 ± 7.4 ^a^	88.3 ± 3.1 ^a^
2	79.5 ± 6.5 ^b^	45.1 ± 2.2 ^b^
3	74.3 ± 10.5 ^b^	40 ± 0.1 ^b^
OP-W ^2^		
1	82.5 ± 4.5 ^a^	44.8 ± 0.8 ^a^
2	68.9 ± 3.0 ^b^	22.7 ± 0.6 ^b^
3	73.6 ± 2.2 ^b^	14.4 ± 1.0 ^c^

^1^ OP-W n.a. (samples 1, 2 and 3): not absorbable aqueous extracts (m.w. > 3.5 kDa). ^2^ OP-W (samples 1, 2 and 3): absorbable aqueous extracts (m.w. < 3.5 kDa). ^a,b,c,^: different letters mean significant differences at *p* < 0.05.

**Table 6 molecules-28-02122-t006:** The radical scavenging activity of the seven fractions obtained from OP-W1, the aqueous absorbable OP extract (OP-F) characterized by the highest antioxidant properties. Values are reported as Trolox equivalent (μg TE/mg dry extract).

	ABTS	TPC
OP-F	Trolox Equivalent ± SD	mg of GAE/g Extract ± SD
1	63.3 ± 5.3 ^AB^	156.7 ± 0.7 ^A^
2	26.5 ± 0.6 ^A^	127.3 ± 3.1 ^A^
3	369.7 ± 4.0 ^C^	719.5 ± 10.9 ^B^
4	342.2 ± 5.8 ^C^	761.8 ± 10.0 ^B^
5	1036.7 ± 35.5 ^D^	1199.4 ± 69.9 ^C^
6	76.5 ± 3.2 ^BE^	197.4 ± 9.7 ^A^
7	253.7 ± 8.3 ^F^	450.5 ± 28.6 ^D^

Different capital letters mean significant differences; *p* < 0.0001.

**Table 7 molecules-28-02122-t007:** List of identified OP-W1 peptides and corresponding proteins by LC-MS/MS analysis. The databases consulted were FASTA-file Olea and *Olea europaea* (common olive).

OP-W1 Proteins				
ID	Protein	Organism	MW	Peptides
J9XLG0	Putative polyphenol oxidase	*Olea europaea*	53658	2
E3TJS3	50S ribosomal protein L16	*Olea europaea*	15346	2
A0A0N9LRR6	Amine oxidase	*Olea europaea*	87207	4
B2VPR8	Pectin esterase 2	*Olea europaea*	39856	2
A0A0G3FBC7	LIP (fragment)	*Olea europaea*	11938	2
A4GE45	Profilin-1	*Olea europaea*	14520	2
A0A649ZUF2	4-coumarate-CoA ligase	*Olea europaea*	59898	3
Q5DTB7	Ole e 3 allergen	*Olea europaea*	5795	2
J9XH65	Putative geraniol 10-hydroxylase	*Olea europaea*	46724	5
A0A126X2X6	Putative LOV domain-containing	*Olea europaea*	70139	2
A0A7G7YFM0	Ribulose bisphosphate carboxylase	*Olea europaea*	53072	2
Q1W4C7	Hexosyltransferase	*Olea europaea*	31071	2
J9XLE5	Isopentenyl-diphospate Delta-isomerase	*Olea europaea*	25822	2
A0A1B1V5C3	Putative transcriptional corepressor	*Olea europaea*	54598	2

**Table 8 molecules-28-02122-t008:** List of identified peptides and corresponding proteins from organisms in the OP-W1 sample by LC-MS/MS analysis. The databases consulted were FASTA-file Olea and Olea europaea (common olive).

Proteins from Organisms in OP-W1 Sample				
ID	Protein	Organism	MW	Peptides
A0A2A2DN42	Chemotaxis protein	*Pseudomonas* sp. *PIC141*	57171	2
A0A2A2DPP3	Short chain dehydrogenase	*Pseudomonas* sp. *PIC 125*	28887	3
A0A2A2E801	Poly(A) polymerase	*Pseudomonas* sp. *PIC 125*	53512	2
A0A2A2DPY9	PhoH family protein	*Pseudomonas* sp. *PIC 141*	38446	2
A0A2A2E7V3	UvrABC system protein C	*Pseudomonas* sp. *PIC125*	67237	2
A0A2A2E543	Amine oxidase	*Pseudomonas* sp. *PIC141*	62484	2
A0A2A2DX44	Serine hydrolase	*Pseudomonas* sp. *PIC141*	40686	2
A0A2A2DU16	Haemagglutinin	*Pseudomonas* sp. *PIC125.*	9783	2
A0A2A2DPN3	DUF1329 domain	*Pseudomonas* sp. *PIC125*	50476	2
A0A2A2DJ68	TonB-dep. siderophore receptor	*Pseudomonas* sp. *PIC125*	78117	3
A0A2A2EB40	Tail-specific protease	*Pseudomonas* sp. *PIC125*	77756	3
A0A2A2E740	RNA helicase	*Pseudomonas* sp. *PIC125*	48800	3
A0A2A2E9Q7	Coproporphyrinogen-III oxidase	*Pseudomonas* sp. *PIC125*	53148	2

**Table 9 molecules-28-02122-t009:** List of identified metabolites (expressed in percentage) recognized in the fifth OP-F sample by GCMS analysis.

Metabolites	%
Trietanolamine	3.07
Propanamine,N(2fluorophenyl)3(4morpholyl)	0.18
Glycerol	3.2
Ciclopenthylamine	2.26
1 propanamine N,N diproyl	2.26
Cystathyonine	1.04
Tris,N-acetyl	1.08
Homocisteine	7.1
Tyrosol	20.49
Glutamic acid	7.61
4 Hexylphenol	4.52
Phenol,3 butanol	0.58
Bis oxyethylthiosulfide	2.74
Linoleic acid	0.23
Sugar	3.25

**Table 10 molecules-28-02122-t010:** List of primers (Sigma-Aldrich Life Science Co. LLC., U.S.A.) utilized in the present study.

Gene	5′-Forward-3′	5′-Reverse-3′
IL-6	GCAGAAAAAGGCAAAGAATC	CTACATTTGCCGAAGAGC
IL-10	GCCTTTAATAAGCTCCAAGAG	ATCTTCATTGTCATGTAGGC
TNF-α	AGGCAGTCAGATCATCTTC	TTATCTCTCAGCTCCACG
18S	GTAACCCGTTGAACCCCATT	CCATCCAATCGGTAGTAGCG

## Data Availability

Data supporting the results of this study are available from authors and are available on request.

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
