# Peer review of "Characterization and Biological Activities of In Vitro Digested Olive Pomace Polyphenols Evaluated on Ex Vivo Human Immune Blood Cells"

_molecules, 2023, doi:10.3390/molecules28052122_

Round 1

Reviewer 1 Report

Though it has well been investigated that olive-oil residue possibly was more useful than of olive oil itself as a source of physiologically active materials, the results shown in the figure 2 and 3 exhibit that the OP could be novel anti-inflammatory resource. 

The referee ask the authors what these peaks 1 to 7 are. In this paper, only peak 5 is explained as tyrosol, however. The peak 3,4 and 7 showed not a small anti-oxidant activity, too.  

And the referee can not understand why the authors does not show what they are. Though the authors can identify those proteins using LC MS/MS!

Author Response

Dear Referee,

We would thank you for having dedicated your time to the critical reading of the manuscript and gave useful suggestion for further studies.

Below, the reply to the query formulated.

Best regards,

Daniela Beghelli

Reviewer 1.

Though it has well been investigated that olive-oil residue possibly was more useful than of olive oil itself as a source of physiologically active materials, the results shown in the figures 2 and 3 exhibit that the OP could be novel anti-inflammatory resource. 

The referee asks the authors what these peaks 1 to 7 are. In this paper, only peak 5 is explained as tyrosol, however. The peak 3,4 and 7 showed not a small antioxidant activity, too.

And the referee cannot understand why the authors does not show what they are. Though the authors can identify those proteins using LC MS/MS!

The authors agree with the refer that it would be very interesting to also characterize the fractions 3,4, and 7. However, the approach that has been followed was a so-called “bio guided-assay” (Weller, 2012; https://doi.org/10.3390/s120709181), which consists in the selection of the biologically active fraction obtained from a complex mixture for a more detailed chemical analysis and characterization. Since fraction 5 showed an antioxidant capacity 3-fold higher than fractions 3, 4 and 7, it was selected for further characterization and analysis.

Besides this, the metabolomic analysis was made ‘in outsourcing service’ (at CEINGE-Biotecnologie Avanzate s.c. a r.l., Naples, Italy web site: www.ceinge.unina.it), and, on the base of the available economic resources, Authors were not able to sustain the ‘effort’ for the four samples.

Undoubtedly, however, the remaining fractions (stored in the freezer) are worthy of further investigation that will be carried out as soon in future studies.

Reviewer 2 Report

The results that are presented in this paper give a good idea on the chemical characterization, antioxidant properties, bio-accessibility after in vitro digestion as well as some anti-inflammatory effects on human mononuclear cells of olive pomace extracts.  This paper is interesting but, nevertheless, I have severe methodological objections, which prevents its publication in its current form.

·        No details are given on the 3 batches of sun-dried OP that were used and it is however mandatory to know more on their production; otherwise, for the readers, there is almost no interest to compare them.  If, as indicated by the authors, agro-pedoclimatic, storage conditions, sun exposure, … are factors involved in the differences that they report, they should be specified to allow reproducibility and a better knowledge of their influence, both at a scientific and industrial levels.

·        The authors totally bypass the crucial question of the bioavailability of their extracts, which, obviously, is of crucial importance.  No experimental or even literature-based information on the bioavailability of the compounds present in the selected extract, which is crucial to evaluate the concentration that will be in contact with the cells in the bioactivity testing.

·        What is indeed – and as discussed by the authors – the real interest to identify the proteins present in the extracts for a product that should be used orally?  Additionally, no data are presented about the presence of other micronutrients such as, i.a. minerals.

·        The distinction between “absorbable” and “not absorbable” fractions is partially artificial due to the lack of “real” intestinal step, which should include a possible involvement of the brush border enzymes in the digestion of the “not absorbable” extract as well as the transport systems in intestinal absorption.

·        The unphysiological presence of oxygen during gastric and intestinal digestion could affect the process.

·        The pharmacological significance of the concentration of 2.5 µg/ml to evaluate the anti-inflammatory activities should be discussed.  More precise values should be given for the concentrations of the bioactive molecules and compared to those that could be reached in “pharmacological” conditions.

·        The valorization of vegetal by-products is obviously a sustainable and economically interesting project.  Nevertheless, these items are not discussed in the paper.

Author Response

Dear Reviewer,

We would thank you for having dedicated your time to the critical reading of the manuscript, but also for giving us the opportunity to improve the manuscript with useful suggestions.

We have addressed the raised aspects and we have modified the manuscript accordingly.

Below are listed, point by point, the replies to the queries formulated and, in the manuscript, it is possible to track the details of the revisions to the manuscript.

Best regards,

Daniela Beghelli

Comments and Suggestions for Authors

The results that are presented in this paper give a good idea on the chemical characterization, antioxidant properties, bio-accessibility after in vitro digestion as well as some anti-inflammatory effects on human mononuclear cells of olive pomace extracts.  This paper is interesting but, nevertheless, I have severe methodological objections, which prevents its publication in its current form.

  • No details are given on the 3 batches of sun-dried OP that were used and it is however mandatory to know more on their production; otherwise, for the readers, there is almost no interest to compare them.  If, as indicated by the authors, agro-pedoclimatic, storage conditions, sun exposure, … are factors involved in the differences that they report, they should be specified to allow reproducibility and a better knowledge of their influence, both at a scientific and industrial levels.

Authors thank reviewer for this suggestion and more information on the origin of the different OP batches, together with a possible explanation of their different phenolic contents, have been added in the text.

Olive pomace samples were collected and stored in three different periods: OP1 in September, OP2 in October and OP 3 in November/December, thus the ripening state was increasing during the harvesting olives. These OP samples were stratified (and stabilized with the Consoli ‘s patent), one batch on top of the other, in a unique big pool, and stabilized with the Consoli‘s patent (Consoli’s patent 0001428707). The pool was well covered till the following summer season, when the pool was open, and OP dried under the sun. Olive pomace samples were collected from the bottom of the pool at the end of June (OP1), of July (OP2), and August (OP3), respectively.

This information was added to paragraph 4.2. Olive pomace methanolic extracts.

  • The authors totally bypass the crucial question of the bioavailability of their extracts, which, obviously, is of crucial importance.  No experimental or even literature-based information on the bioavailability of the compounds present in the selected extract, which is crucial to evaluate the concentration that will be in contact with the cells in the bioactivity testing.

IN agreement with the reviewer’s suggestion, authors  introduced in the text some recent references and cited a very interesting article (Sakavitsi et al., 2022; https://doi.org/10.3390/metabo12050391) on the bioavailability of the main compounds (olive phenolic alcohols Hydroxytyrosol and Tyrosol ) present in the selected extracts utilized on PBMC. Based on these additions, the discussion was revisited and integrated.

  • What is indeed – and as discussed by the authors – the real interest to identify the proteins present in the extracts for a product that should be used orally? Additionally, no data are presented about the presence of other micronutrients such as, i.a.minerals.

Authors have modified the sentence at line 420-421.

Indeed, Authors intended ‘peptides’ instead of proteins that, obviously were in vitro digested.

Furthermore, stimulated by Reviewer (thank you, again), authors deeply analyzed the identified peptides in the PeptideRanker data base and then, the potentially bioactive peptides were checked in BIOPEP-UWM Database of Bioactive Peptides. The text was accordingly revised also in the Results and Material and Methods paragraphs.

  • The distinction between “absorbable” and “not absorbable” fractions is partially artificial due to the lack of “real” intestinal step, which should include a possible involvement of the brush border enzymes in the digestion of the “not absorbable” extract as well as the transport systems in intestinal absorption.

The distinction between “absorbable” and “not absorbable” fractions has been inspired by the papers of Ribeiro et al. (2018, 2020, 2021).

Regarding ‘a possible involvement of the brush border enzymes on the “not absorbable” extract’, the authors fully agree with the question raised by the reviewer.

All models for cause/effect study are simpler than reality, but obviously less 'reliable' of the physiological situation in vivo. Therefore, as evaluating extracts on cells directly cannot produce true and accurate results, likewise the in vitro simulation of digestive process cannot be 100% reliable.

In this in vitro simulation of the digestion process it is not possible, in fact, to reproduce neither the action tied to the presence of the small intestine brush border (and of its enzymes), neither the action of gut microbiota that is fully skipped (but could further impact on the availability of the “non absorbable fraction”). Furthermore, also the endogenous production of hydroxytyrosol and tyrosl from dopamine and tiramine, respectively, could be not tested (Nikou et al.; Nutrients 2022, 14(18), 3773; https://doi.org/10.3390/nu14183773) in the present in vitro model.

By using in vitro digestion, however, researchers are perhaps closer to the truth than by directly using the extracts in cell-based models.

On the other hand, these in vitro models present the advantages that they are not influenced by age, sex, dietary habits, microbiome composition, genetic variation, drug exposure etc..

These considerations have been included in the Conclusions section as a limit of this study.

The unphysiological presence of oxygen during gastric and intestinal digestion could affect the process.

Authors agree with the reviewer that, undoubtedly, the presence of oxygen during the simulation of the digestive process is representative of an unphysiological situation. Nevertheless, authors think the presence of oxygen could have exerted a pejorative effect on the intrinsic antioxidant properties of phenols (by oxidating them) present in the olive pomace.

As previously stated, all the in vitro study models are imperfects, and it is really very hard/impossible to fully replicate in vitro what happens in vivo.

Even if Sakavitsi et al. 2022 were able to utilize an advanced in vitro human GI digestion model (In Vitro GIDM–Colon Model to digest pure hydrytyrosol and tyrosol, these authors affirmed that using “an in vitro model that mimics the human GI tract could lead to significant new insights into the biotransformation reactions after oral ingestion" represents ‘a privilege’ for research. These authors described a semi-anaerobic condition for the small intestine.

The in vitro model of simulated digestion used by authors in the present work was also used by other authors:

Diab et al., 2022; doi:10.3390/antiox11091778.

Ribeiro et al., 2020; doi:10.1039/C9FO03000J.

Ribeiro et al., 2021; https://doi.org/10.1016/j.foodres.2020.110032

Minekus, et al., 2014; https://doi.org/10.1039/C3FO60702J

  • The pharmacological significance of the concentration of 2.5 µg/ml to evaluate the anti-inflammatory activities should be discussed. More precise values should be given for the concentrations of the bioactive molecules and compared to those that could be reached in “pharmacological” conditions.

According to the reviewer suggestion, the authors have specified in the text the concentrations of the bioactive molecules.

Here authors reported some data relative to the pharmacokinetic of NSAID like: diclofenac, ibubrofen, and aceclofenac. Successively the documented bioavailability of hydroxytyrosol and tyrosol -after an oral administration (in humans) - is reported in comparison with the extract concentration adopted in the present work.

Dose suggested for Diclofenac, the highest available anti-inflammatory drug, or ibuprofen are: 100-150 mg/day (blood peak:1,0 mg/ml (for 25 mg), 1,5 mg/ml (50 mg), 2,0 mg/ml (75 mg)) or 400 mg/day (blood peak 36-39 mg/ml) respectively; whereas, in the study of Bushra et al.,(DOI: 10.1371/journal.pone.0238951), it was compared the ‘the various pharmacokinetic parameters of the newly developed cost-effective aceclofenac 100 mg tablet formulation vs against the marketed brand (ACEMED) to establish the bioequivalence’...

In the aforementioned study, Aceclofenac (100 mg) optimized formulation and the marketed (ACEMED) tablets were administered orally. Authors observed a plasma peak of about 9 microg/ml at 2 h.

In the present study, the extract concentration directly utilized on ex vivo cells was 2.5 mg/ml that correspond to a concentration of 2.17ng/ml of hydroxytyrosl or 0.0035 ng/ml of tyrosol, 0.75 ng/ml of oleuropein, 0.9 ng/ml of myricetin, and 0.95 ng/ml of luteolin. Our main objective was to use concentrations reachable in plasma after oral consumption of food containing these compounds.

In particular, Kountouri et al. observed that the oral administration of olives (approximately 100 g) containing hydroxytyrosol and tyrosl (76.73 and 19.48 mg/100 g olives, respectively) gave the following plasma concentration of tyrosol (1-4hrs):

Whereas, González-Santiago, by orally administering hydroxytyrosl 2.5mg/kg b.w., in human volunteers, obtained a concentration of 1.11micromol/L (170microg/L).

Authors believe that the preliminary results observed in the present study about the anti-inflammatory properties of OP-W are promising. The low dosage at which it resulted effective compared with the ones of a NSAID molecule after oral assumption allows to say that the intake of even just 100 grams of olives could have an anti-inflammatory effect, although transitory, that is repeated whenever you introduce, with the diet, the biomolecules present in the olive and its waste products.

These are preliminary results, however, require further investigations.

The valorization of vegetal by-products is obviously a sustainable and economically interesting project.  Nevertheless, these items are not discussed in the paper.

In accordance with the suggestions of all the Reviewers, Authors have extended the paragraph of Conclusions summarizing the results obtained in the present research, discussing the importance of the valorization of vegetal by-products in circular economy, but also adding the limitations of the study observed by reviewers. These limitations have been discussed as possible new perspectives of investigation.

Reviewer 3 Report

Respected Authors,

The topic of article is interesting and important. Olive pomace seems good source of high quantities of health-promoting bioactive compounds.  Therefore, the undertaken research topic by Authors is essential. The aim of article was good presented. The authors explained their choice of cell line to study. Results of conducted study were good described. I am under impression of characterization of three batches of sun-dried olive pomace. The conclusions should be extended and summarized better results, which have been discussed in detail in relation to other studies.

The reviewer suggests minor revisions. The list of suggestions and remarks are below:

11.       Authors should standardize the spelling TNF-α (for example line 396 and 413). I suggest TNF-α

22.       Below Table 6 Authors should explain meaning of different letters

33.       In line 258 Authors should specify type of TNF

44.       In part 4.2.3 (line 501) – Authors should enter a subscript

55.       In line 568 Authors should remove dot.

66.       In line 620 Authors should correct word colture, I suggest: culture

77.       Conclusions should be extended and summarize the obtained results

Author Response

REVIEWER 3

Dear Reviewer,

We would thank you for having dedicated your time to the critical reading of the manuscript, but also for giving us the opportunity to improve the manuscript with useful suggestions.

We have addressed all the raised aspects and we have modified the manuscript accordingly.

Below are listed, point by point, the replies to the queries formulated and, in the manuscript, it is possible to track the details of the revisions to the manuscript.

Best regards,

Daniela Beghelli

Comments and Suggestions for Authors

Respected Authors,

The topic of article is interesting and important. Olive pomace seems good source of high quantities of health-promoting bioactive compounds. Therefore, the undertaken research topic by Authors is essential. The aim of article was good presented. The authors explained their choice of cell line to study. Results of conducted study were good described.

I am under impression of characterization of three batches of sun-dried olive pomace.

In accordance with the reviewer’s suggestions, Authors have in the text better explained the differences existing between the three olive pomace batches.

In the intention of the authors, it was wanted to start from a number of samples greater than 1 to have the opportunity to choose the most promising lot to be characterized gradually more and more in depth according to the biological activity.

Olive pomace samples were collected and stored in three different periods: OP1 in September, OP2 in October and OP 3 in November/December, thus the ripening state was increasing during the harvesting olives. These OP samples were stratified (and stabilized with the Consoli ‘s patent), one batch on top of the other, in a unique big pool, and stabilized with the Consoli‘s patent (Consoli’s patent 0001428707). The pool was well covered till the following summer season, when the pool was open, and OP dried under the sun. Olive pomace samples were collected from the bottom of the pool at the end of June (OP1), of July (OP2), and August (OP3), respectively.

The conclusions should be extended and summarized better results, which have been discussed in detail in relation to other studies.

(77.       Conclusions should be extended and summarize the obtained results)
In accordance with the suggestions of all the Reviewers, Authors have extended the paragraph of Conclusions summarizing the results obtained in the present research, but also adding the limitations of the study observed by reviewers. These limitations have been discussed as possible new perspectives of investigation.

The reviewer suggests minor revisions. The list of suggestions and remarks are below:

  1. Authors should standardize the spelling TNF-α (for example line 396 and 413). I suggest TNF-α

The authors have addressed the suggestion and have corrected the spelling in TNF-α.

  1. Below Table 6 Authors should explain meaning of different letters

Under Table 6, authors have written “A,B,C,..Different letters mean significant differences : p<0.0001”.

  1. In line 258 Authors should specify type of TNF

Thank you. The authors have corrected the spelling in TNF-α.

  1. In part 4.2.3 (line 501) – Authors should enter a subscript

Thank you. The authors have inserted the required subscript.

  1. In line 568 Authors should remove dot.

Thank you. The authors have removed the dot.

  1. In line 620 Authors should correct word colture, I suggest: culture

Thank you! The authors have corrected the spelling.

  1. Conclusions should be extended and summarize the obtained results
    In accordance with the suggestions of all the Reviewers, Authors have extended the paragraph of Conclusions summarizing the results obtained in the present research, but also adding the limitations of the study observed by reviewers. These limitations have been discussed as possible new perspectives of investigation.

Round 2

Reviewer 2 Report

Significant improvment of this revised version that largely - not entirely - answers to my previous objections.